# Enzymatic Synthesis of Vancomycin-Modified DNA

**DOI:** 10.3390/molecules27248927

**Published:** 2022-12-15

**Authors:** Chiara Figazzolo, Frédéric Bonhomme, Saidbakhrom Saidjalolov, Mélanie Ethève-Quelquejeu, Marcel Hollenstein

**Affiliations:** 1Institut Pasteur, Université de Paris Cité, CNRS UMR3523, Department of Structural Biology and Chemistry, Laboratory for Bioorganic Chemistry of Nucleic Acids, 28, rue du Docteur Roux, CEDEX 15, 75724 Paris, France; 2Learning Planet Institute, 8, Rue Charles V, 75004 Paris, France; 3Institut Pasteur, Université de Paris Cité, Department of Structural Biology and Chemistry, Unité de Chimie Biologique Epigénétique, UMR CNRS 3523, 28 rue du Docteur Roux, CEDEX 15, 75724 Paris, France; 4Université Paris Cité, CNRS, Laboratoire de Chimie et Biochimie Pharmacologiques et Toxicologiques, 75006 Paris, France

**Keywords:** modified nucleic acids, DNA polymerases, nucleoside triphosphates, primer extension reactions, SELEX, antimicrobial resistance

## Abstract

Many potent antibiotics fail to treat bacterial infections due to emergence of drug-resistant strains. This surge of antimicrobial resistance (AMR) calls in for the development of alternative strategies and methods for the development of drugs with restored bactericidal activities. In this context, we surmised that identifying aptamers using nucleotides connected to antibiotics will lead to chemically modified aptameric species capable of restoring the original binding activity of the drugs and hence produce active antibiotic species that could be used to combat AMR. Here, we report the synthesis of a modified nucleoside triphosphate equipped with a vancomycin moiety on the nucleobase. We demonstrate that this nucleotide analogue is suitable for polymerase-mediated synthesis of modified DNA and, importantly, highlight its compatibility with the SELEX methodology. These results pave the way for bacterial-SELEX for the identification of vancomycin-modified aptamers.

## 1. Introduction

Bacterial antimicrobial resistance (AMR) is one of the biggest public health challenges of the 21st century and occurs when bacteria develop mechanisms that protect them against the effects of antibiotics [1]. AMR can be intrinsic to bacteria, or can be acquired through horizontal gene transfer or mutations in chromosomal genes [2]. This major threat to human health has caused 4.95 million deaths around the world in 2019, and these numbers are expected to rise steeply in the near future [3]. Consequently, an important first step in combatting AMR consists of refining our understanding of the mechanisms underlying drug failures [4,5]. This knowledge will, in turn, guide the development of alternative antimicrobial drugs as well as that of unconventional methods for drug design [6,7,8,9,10]. In this context, vancomycin is amongst the most clinically relevant and effective glycopeptide antibiotics used for the treatment of infections caused by Gram-positive pathogens [11,12]. This antibiotic is considered as a drug of last resort due to its high efficiency, particularly against methicillin-resistant *Staphylococcus aureus* (MRSA) and *Staphylococcus epidermidis* (MRSE) [13]. The mechanistic details of vancomycin-driven bactericidal activity have been resolved and involve tight binding to a terminal section of the precursor of the polymer peptidoglycan on the bacterial cell surface. This binding event further blocks transglycosylation and transpeptidation, which are essential for cell wall synthesis and maintenance (Figure 1) [11,12,14]. The efficiency and popularity of vancomycin has led to a clinical overuse [15] of this glycopeptide antibiotic which explains the emergence of resistance in bacterial pathogens. Vancomycin resistance is quite complex, but can essentially either be acquired by target modification or be intrinsic [12,16]. Acquired vancomycin resistance essentially occurs via a simple change in the amino acid composition of the peptidoglycan from D-Ala-D-Ala to D-Ala-D-Lac (X = NH and O in Figure 1, respectively). This minute alteration dramatically reduces the affinity of vancomycin for its peptidoglycan target which causes a significant loss in antimicrobial activity. Intrinsic resistance mechanisms include biofilm construction and formation of dormant stationary phase subpopulations [12,16,17,18]. In general, strategies implemented to combat vancomycin resistance and AMR strive to develop new, small molecule drugs [19,20,21], modify existing scaffolds to extend their lifetimes [13,22,23,24], or explore alternative routes such as the introduction of the multivalency concept [25,26,27] or prevention of biofilm formation [12,28]. Despite significant success of these approaches, universal and robust strategies to combat AMR are still in dire need. Herein, we present a first step towards the development of a novel strategy that combines aptamer recognition, vancomycin analogues tethered to DNA nucleobases, and bacterial-SELEX. Towards this aim, we have synthesized a nucleoside triphosphate equipped with a vancomycin moiety at the C5 site of the nucleobase. We then demonstrated the compatibility of this modified nucleotide with enzymatic DNA synthesis and with the SELEX methodology.

## 2. Results and Discussion

### 2.1. Design and Chemical Synthesis of Vancomycin-Modified Nucleoside Triphosphate (dU**^Van^**TP)

Aptamers are single-stranded nucleic acids capable of binding to targets with high affinity and specificity [31,32]. The first applications of aptamers in the context of microbial infections have previously been reported [33] including inhibition of biofilm formation, delivery of antibiotics conjugated to their scaffold, and participation in specific pathogen destruction [34,35,36]. In contrast, only a few reports have been dedicated to the application of aptamers to combat AMR, mainly as drug delivery systems [37]. For instance, aptamers conjugated to photosensitizers have been proposed as vectors for photodynamic therapy (PDT) to treat infections caused by antibiotic resistant pathogens [38,39,40]. In these examples, previously selected aptamers specific for particular bacterial targets are converted by post-SELEX modification to specific drug delivery agents. While this approach has met some success, post-SELEX modification protocols come with a lot of drawbacks [41,42]. First, the appendage of one or multiple drugs or reporter molecules on the termini of aptamers can lead to a loss in binding affinity (increase in *K*_D_ values) which is detrimental for such an approach. Secondly, most aptamers are often directed towards diagnostic detection purposes rather than as therapeutic agents and hence do not present a sufficiently developed modification pattern robust enough to resist nuclease-mediated degradation. To combat this, the implementation of a post-SELEX modification strategy to mitigate this limitation and improve their biostability will be required. Third, post-SELEX approaches only allow for the introduction of a limited number of antibiotics (mainly by connection via the 3′- and 5′-termini) which induces high-cost productions and relatively low topical drug concentration. Lastly, upon reaching its intended target, the drug will be released but unless this released compound is a new chemical analogue of the parent drug it is unlikely to elude the defense and resistance mechanisms of the pathogens. In light of these elements, we rationalized that the inclusion of antibiotics on the scaffold of nucleoside triphosphates will yield ways to generate modified aptamers specific for resistant bacterial pathogens and potentially restore the intended antimicrobial activity. By inspection of the vancomycin scaffold, we rationalized that the free carboxylic acid might be a favorable site for connection to a nucleotide. This moiety is not involved in target recognition and previously reported modification at this site usually yielded analogues that retain the original antimicrobial activity [13,27,30]. To form the connection between vancomycin and the aptamer we opted to use the copper(I)-catalyzed azide–alkyne cycloaddition (CuAAC) reaction, since this chemistry has previously been used to generate modified vancomycin derivatives [13]. In addition to this, nucleotides bearing a triazole-connecting linker arm are compatible with enzymatic DNA synthesis [43,44] and aptamer selection [45,46,47]. The synthesis of the vancomycin-bearing deoxyuridine triphosphate dU**^Van^**TP (Figure 2) involved synthesis of the previously reported azide-modified vancomycin [13] and CuAAC reaction with 5-ethynyl-dUTP (EdUTP) [48] under standard conditions (see Materials and Methods (Section 4.1) and Appendix A for full synthetic details). It is worth mentioning that the application of standard CuAAC reaction conditions yielded a mixture of species, including the expected nucleotide as well as a product corresponding to dU**^Van^**TP coordinated to Cu^2+^ (Appendix A). Coordination of the divalent metal cation occurs preferentially through the *N*-terminal imino NH-CH_3_ unit, two consecutive nitrogen atoms in the peptide chain, and one oxygen atom from the asparagine amide moiety [49,50]. The binding of vancomycin to Cu^2+^ is particularly efficient [50] and, hence, required the implementation of a thorough purification step to remove this undesired, cytotoxic transition metal cation from the modified nucleotide (see Materials and Methods (Section 4.1)). Alternatively, copper-free click chemistry may be considered in the future to avoid this rigorous purification step [51].

### 2.2. Biochemical Characterization of the Modified Nucleotide dU**^Van^**TP

With dU**^Van^**TP in hand, we evaluated the possibility of using this analog to synthesize vancomycin-modified DNA via primer extension (PEX) reactions and PCR. To do so, we first carried PEX reactions using the 20-nucleotide long template **T1** and the 5′-FAM-labelled, 19-mer primer **P1** (Table 1). This rather simple system allows for the incorporation of a single, modified nucleotide and can be followed by gel electrophoresis, LC-MS analysis, and UV-melting experiments. PEX reactions were performed with Vent (*exo*^−^) DNA polymerase (vide infra) and the products were analyzed by gel electrophoresis (Appendix A). This analysis revealed the formation of a product with a slower electrophoretic mobility compared to that of the product obtained with dTTP or the unmodified, single-stranded DNA template. In order to confirm the incorporation of a dU**^Van^** nucleotide into DNA, we subjected the PEX reaction product to a digestion-LC-MS analysis protocol (see Materials and Methods (Section 4.2)) [52]. For the preparation of a standard for the modified nucleoside which is required for such an analysis, we dephosphorylated dU**^Van^**TP with Shrimp Alkaline Phosphatase (rSAP) and analyzed the resulting nucleoside by LC-MS (Figure 3 and Appendix A) and HRMS (Appendix A). PEX reaction products were nuclease-digested, dephosphorylated and the resulting deoxynucleosides were then analyzed by LC-MS (Figure 3).

The LC-MS chromatogram of the digested PEX reaction product using only dA/G/T/CTP displayed the expected four peaks corresponding to each of the four canonical nucleosides (Figure 3A). Similarly, the four canonical nucleosides were also detected in the LC-MS chromatogram of the digested reaction product obtained with dU**^Van^**TP, however, an additional peak at longer retention time (Figure 3B) and with a mass corresponding to dU**^Van^** was also detected (Appendix A and Figure 3C). Therefore this analysis univocally confirmed the presence of the modified dU**^Van^** nucleotide in the extended primer **P1** obtained after PEX reaction.

Next, we sought to evaluate the effect of such a bulky modification on duplex stability. To do so, we performed large-scale PEX reactions on the aforementioned **P1**/**T1** primer/template duplex in the presence of dU**^Van^**TP and Vent (*exo*^−^) DNA polymerase [53,54]. The UV-melting curves of the resulting modified dsDNA were then measured under standard saline buffer conditions and analyzed using a newly written python script for *T*_m_ data analysis (see Figure 4 and Appendix A) [55]. The presence of the vancomycin moiety led to a rather significant destabilization of the duplex (Δ*T*_m_ = −3.9 °C) compared to that of the unmodified dsDNA (62.7 ± 0.3 °C compared to 66.6 ± 0.1 °C). This destabilization might be caused either by the presence of the bulky vancomycin moiety since other large moieties such as polyethylene glycol [56] induced similar decreases in *T*_m_ values, or by the thermal penalty imparted by the introduction of a single triazole-containing nucleotide [57,58,59]. Nonetheless, this UV-melting analysis indicates that the modification did not significantly interfere with duplex formation.

Next, we sought to evaluate the possibility of using dU**^Van^**TP to synthesize longer vancomycin-containing DNA fragments using polymerase-mediated catalysis. To do so, we first carried out PEX reactions using the 19 nucleotide long 5′-FAM-labelled primer (**P2**) along with the corresponding 79-mer template (**T2**) that permits the incorporation of up to nine modified nucleotides (Table 1). A variety of polymerases were employed for PEX reactions, including the Klenow fragment of *E. coli* (Kf *exo*^−^) DNA polymerase I, *Bst*, Hemo Klen Taq, and Taq (family A polymerases), the family B polymerases Phusion, Therminator, Vent (*exo*^−^), phi29, and Deep Vent, and the Y-family polymerase *Sulfolobus* DNA Polymerase IV (Dpo4). Analysis by gel electrophoresis of the resulting PEX reaction products (Figure 5), revealed that a number of polymerases (Phusion, Q5, Therminator, Vent (*exo*^−^) and Deep Vent) readily accepted dU**^Van^**TP as a substrate. By contrast, polymerases such as Taq did not tolerate the modified nucleotide and no extended product was formed. The highest conversion yields were observed when Vent (*exo*^−^) and Therminator were used as polymerases to catalyze the incorporation of the modified nucleotide into DNA. The bands corresponding to full length, modified products possessed a retarded gel electrophoretic mobility compared to that of control reactions performed with canonical nucleotides. This is expected when a bulky residue is attached to the nucleobase [56,60,61]. Despite the rather high substrate tolerance of Therminator for dU**^Van^**TP, this polymerase has been shown to misincorporate natural and modified nucleotides [62,63,64,65], and to display an efficient pyrophosphorolysis capacity followed by incorporation of nucleotides [66]. Consequently, we believe that this catalyst might not be suitable for SELEX applications when used in conjunction with dU**^Van^**TP and instead we used Vent (*exo*^−^) for further experiments.

Having established conditions for the enzymatic preparation of long, vancomycin-containing DNA oligonucleotides via PEX reactions, we next set out to evaluate the compatibility of dU**^Van^**TP with PCR. To do so, we performed PCR with template **T2** along with the forward and reverse primers **P3** and **P4**, respectively. We evaluated the capacity of three thermostable polymerases (i.e., Deep Vent, Vent (*exo*^−^), and Phusion) to amplify template **T2** in the presence of the modified nucleotide, dU**^Van^**TP. Surprisingly, the expected amplicons were produced in rather low yields, suggesting that the modified nucleotide is not very well tolerated by polymerases under PCR conditions (Appendix A). Despite the low synthetic yields, PCR represents an alternative to PEX reactions to produce vancomycin-containing dsDNA.

### 2.3. Compatibility of Vancomycin-Modified Nucleotide with SELEX

After establishing the conditions for the enzymatic synthesis of vancomycin-modified DNA, we next sought to evaluate the compatibility of dU**^Van^**TP with the SELEX method for the identification of aptamers [45,67,68,69,70]. SELEX and combinatorial methods of in vitro evolution with modified nucleotides require (1) synthesis of modified, degenerate libraries using PEX reactions or PCR; (2) robust conversion from double-stranded to single-stranded modified libraries [71] and (3) an efficient and high-fidelity regeneration of unmodified DNA from modified templates [54,72,73,74]. In order to assess whether enzymatic synthesis with dU**^Van^**TP fulfilled these criteria we first performed PEX reactions with a degenerate library (template **T3** in Table 1), a panel of polymerases and dTTP substituted by dU**^Van^**TP. As with a defined sequence of the same length, the Therminator, Vent (*exo*^−^), and Deep Vent DNA polymerases were capable of producing the expected full-length products in high yields (Figure 6). Thus these polymerases could produce fully modified dsDNA libraries and process these more demanding templates in the presence of the bulky modified nucleotide.

Next, we evaluated the possibility of reverse transcribing vancomycin-containing DNA oligonucleotides to unmodified sequences. To do so, we synthesized vancomycin-containing dsDNA by PCR as described above. After a thorough purification step, we subjected the modified product to PCR with canonical nucleotides and Taq (Figure 7A) or Vent (*exo*^−^) (data not shown) as DNA polymerases. Agarose gel electrophoretic analysis of the resulting reaction products clearly indicated the formation of the expected amplicon (Figure 7A). We next subjected the obtained PCR product with canonical nucleotides and the Taq polymerase to a TA-cloning step followed by Sanger sequencing of nine individual colonies. Sequencing of this small subset of colonies clearly indicated the absence of mutations compared to the expected parent sequence of template **T2** (Figure 7B). This analysis suggested a faithful reverse transcription of vancomycin-containing DNA to unmodified oligonucleotides. Lastly, we evaluated the possibility of converting double-stranded, modified sequences to single-stranded oligonucleotides, which is an important prerequisite for SELEX and a notoriously difficult step, especially for heavily modified sequences or bulky modifications such as vancomycin [71]. We first performed PEX reactions with dU**^Van^**TP, Vent (*exo*^−^), and a 5′-phosphorylated analogue of template **T1** (template **T4**, Table 1). The resulting modified dsDNA was then converted to the corresponding ssDNA by λ-exonuclease digestion (Appendix A). Gel analysis of the resulting products indicated formation of the expected single-stranded sequences (Appendix A).

## 3. Conclusions

AMR is a major threat to human health, and alternative treatment modalities and drugs are in dire need. Herein, we have presented the first step towards combining antibiotics and aptamers in order to potentially restore the effectiveness of drugs against resistant bacteria. Towards this aim, we have synthesized a deoxyuridine analogue bearing a vancomycin residue at position C5 of the nucleobase. The resulting nucleotide, dU**^Van^**TP, was then shown to be compatible with enzymatic DNA synthesis under PEX reaction conditions and PCR (to a certain extent), despite the presence of such a bulky chemical modification. We have then demonstrated the compatibility of dU**^Van^**TP with the SELEX methodology. In particular, we have shown that modified libraries could be generated by enzymatic synthesis and subsequently reverse transcribed into unmodified DNA without introducing mutations. Cell-SELEX experiments against vancomycin resistant strains and subsequent evaluation of the antibacterial activity of modified aptamers are currently underway. We expect the aptamer section of such a construct to bind to membrane and surface proteins, which are the typical targets of cell-SELEX experiments [75]. This binding event will then, in turn, restore binding of vancomycin to the peptidoglycan and, hence, restore its activity. A similar approach using a primer equipped with a hotspot peptide in SELEX was recently employed for the selection of aptamers against the human angiotensin-converting enzyme 2 (hACE2) [76] underscoring the usefulness of such a method. However, the advantage of our approach is that multiple vancomycin-bearing nucleotides will be integrated within the scaffold of the aptamer and will be involved in the binding interactions. Hence, uncertain, labor-intensive post-SELEX structure-activity relationship studies to optimize aptamers will not be required. Lastly, we believe that such an approach could easily be extended to other antibiotics such as β-lactamase inhibitors [7,8] or pentamidine-based scaffolds [77].

## 4. Materials and Methods 

### 4.1. Chemical Synthesis of dU**^Van^**TP

EdUTP [48] (2 mg, 0.004 mmol, 1.5 eq.) and azido-vancomycin [13] (4 mg, 0.0026 mmol, 1 eq.) were respectively dissolved in 100 μL H_2_O each in two different Eppendorf tubes and degassed with argon. In a separated Eppendorf tube, CuI (3 mg, 0.02 mmol, 0.77 eq.) was suspended in DIPEA (10 μL, 0.06 mmol, 2.3 eq.), 50 μL H_2_O and 50 μL CH_3_CN and degassed with Argon. The three fractions were merged and 100 μL DMSO were added. The reaction mixture was degassed with Argon and left to shake at 1000 rpm, at r.t. for 3 h. The reaction mixture was added to 12 mL NaClO_4_ 2% in acetone to precipitate the crude product, centrifuged at 4000 rpm for 15 min, and the surnatant was discarded. The crude was redissolved in 3 mL H_2_O and purified by anion exchange analytical HPLC (100% B in 25 min; Buffer A, 10 mM TEAB in H_2_O; Buffer B, 1 M TEAB in H_2_O) with a DNA Pac PA 100 oligonucleotide column. Prior to each injection, EDTA (0.5 M, pH = 8) was added to the crude solution to coordinate the remaining Cu^2+^ present in the solution. The fractions corresponding to dU**^Van^**TP were merged together and lyophilized overnight to give the product as 2.9 mg (55%) of a white powder. To remove the remaining coordinated Cu^2+^, dU**^Van^**TP was dissolved in 1 mL H_2_O and stirred with 1 g of Chelex 100 resin for 2 h at rt.

^31^P NMR (202.4 MHz, D_2_O): −6.15 (d, *J* = 20.2 Hz, 1P), −11.17 (d, *J* = 18.2 Hz, 1P), −22.17 (t, *J* = 19.2 Hz, 1P); HR-ESI-MS (*m*/*z*): calcd for M = C_79_H_94_Cl_2_N_15_O_37_CuP_3_, [M] = 2071.00, found [M-2H]^2−^ = 2068.3518. HR-ESI-MS (*m*/*z*): calcd for M = C_79_H_94_Cl_2_N_15_O_37_P_3_, [M] = 2009.48, found [M-2H]^2−^ = 2007.4412.

### 4.2. Preparation of dU**^Van^** Nucleoside Standard

dU**^Van^**TP (10 nmol) was incubated with rSAP (10 µL) and rCutSmart buffer (2.2 µL of 10X) in a total volume of 22 µL for 1 h at 37 °C. Afterwards, rSAP was inactivated via incubation for 10 min at 65 °C. The product was purified and characterized via LCMS (Appendix A). HR-ESI-MS (*m*/*z*): calcd for M = C_79_H_91_Cl_2_N_15_O_28_Cu, [M] = 1833.12, found [M]^+^ = 1832.4842. HR-ESI-MS (*m*/*z*): calcd for M = C_79_H_91_Cl_2_N_15_O_28_, [M] = 1769.57, found [M+2H]^2+^ = 1770.5730.

### 4.3. Digestion and LC-MS Analysis of PEX Reaction Product

The primer **P1** (100 pmol) was annealed to the template **T1** (150 pmol) in H_2_O by heating to 95 °C and then gradually cooling to room temperature (over 1 h). Vent (*exo*^−^) polymerase (4U), Thermopol 10X buffer (provided by the supplier of the DNA polymerase) (2 μL) and dU**^Van^**TP (or dTTP for the positive control) (200 μM) were added for a total volume of 10 μL. The reaction mixture was incubated for 4 h at 60 °C. The formation of the products was verified with an agarose E-GEL (4%) using the E-GEL sample loading buffer (1X). The products were purified via Monarch DNA Cleanup Columns (5 μg) (250 pmol product per column). The products (1 μL), the Nucleoside Digestion Mix buffer (2 μL of 10X) and the Nucleoside Digestion Mix (1 μL) were mixed in a total volume of 20 μL and incubated at 37 °C for 1 h. The digested DNA solution was then injected onto a ThermoFisher Hypersil Gold aQ chromatography column (100 × 2.1 mm, 1.9 μm particle size) heated at 30 °C without any further purification. The flow rate was set at 0.3 mL/min and run with an isocratic elution of 1% ACN in H_2_O with 0.1% formic acid for 8 min then 100% ACN from 9 to 11 min. Parent ions were fragmented in positive ion mode with 10% normalized collision energy in parallel-reaction monitoring (PRM) mode. MS2 resolution was 17,500 with an AGC target of 2 × 10^5^, a maximum injection time of 50 ms, and an isolation window of 1.0 *m*/*z*. The inclusion list contained the following masses: dC (228.1), dA (252.1), dG (268.1), dT (243.1) and 4 (884.8; *z* = 2). Extracted ion chromatograms of base fragments (±5 ppm) were used for detection (112.0506 Da for dC; 136.0616 for dA; 152.0565 Da for dG; 127.0501 Da for dT and 1625.4652 Da for 4). Synthetic standards were previously injected to confirm the assignment (fragment ion and retention time).

### 4.4. General Protocol for PEX Reactions

The 5′-FAM-labelled primer (10 pmol) was annealed to the template (15 pmol) in H_2_O by heating to 95 °C and then gradually cooling to room temperature (1 h). The appropriate polymerase, 5X or 10X enzyme buffer (provided by the supplier of the DNA polymerase), the natural nucleotides (200 μM), and dU**^Van^**TP (or dTTP in the positive controls) (200 μM) were added for a total reaction volume of 10 μL. Following incubation for 4 h at the optimal temperature for the enzyme, the reactions were quenched by adding the stop solution (10 μL; formamide (70%), ethylenediaminetetraacetic acid (EDTA; 50 mM), bromophenol (0.1%), xylene cyanol (0.1%)). The reaction mixtures were subjected to gel electrophoresis in denaturing polyacrylamide gel (20%) containing trisborate- EDTA (TBE) 1× buffer (pH 8) and urea (7 M). Products were then analyzed on a phosphorimager.

### 4.5. Thermal Denaturation Experiments

Experiments were recorded in 1X phosphate-buffered saline (PBS) (137 mM NaCl, 2.7 mM KCl, 4.3 mM Na_2_HPO_4_, 1.47 mM KH_2_PO_4_, pH 7.4) in H_2_O with 2 μM final duplex concentration. Paraffin oil (150 μL) was added to avoid evaporation. A control cell (1X PBS) was prepared into which the temperature probe was placed. Absorbance was monitored at 260 nm and the heating rate was set to 1 °C/min. A heating-cooling cycle in the temperature range of 20–95 °C was applied and repeated 3 times per sample. The data analysis was performed with a small Python (v3.8.5) script on the heating ramps (see Appendix A). The absorbance melting curves were converted to hyperchromicities and the first derivative of each experimental hyperchromicity point was calculated. Afterwards, the Gaussian function that best fitted the first derivative curve was calculated and the melting temperature was obtained from it. The final *T*_m_ value was calculated as the mean of the three obtained values with the relative error.

## Figures and Tables

**Figure 1 molecules-27-08927-f001:**
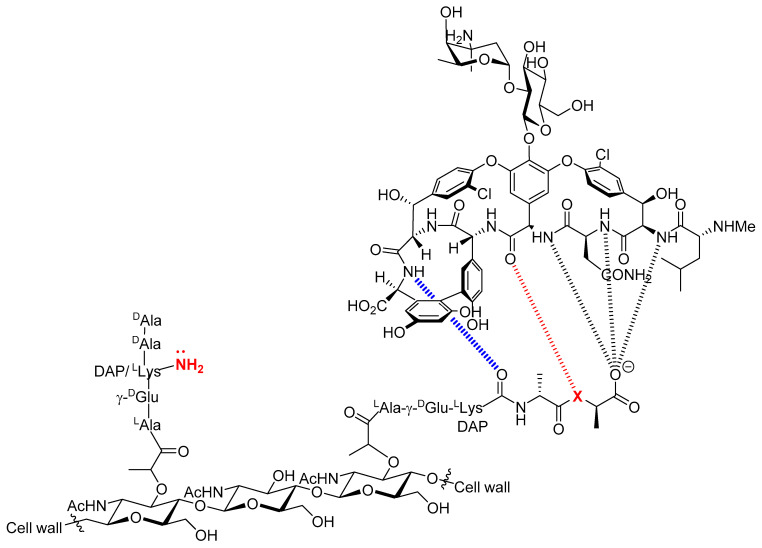
Chemical structure of vancomycin and description of the acquired resistance to vancomycin. The antibacterial activity of vancomycin stems from its high binding affinity(4.4 × 10^5^ M^−1^) for the terminal D-Ala-D-Ala sequence of the precursor peptidoglycan pentapeptide [29]. This binding creates a steric blockade which prevents the enzyme-mediated transpeptidation of the free amine of D-Lys or diamino pimelic acid (DAP) on the second last D-Ala residue from occurring. This in turn induces inhibition of bacterial cell wall biosynthesis. The origin of VanA and VanB bacterial resistance stems from a change in amino acid composition from D-Ala-D-Ala (X = NH) to D-Ala-D-Lac (X = O) [14]. The presence of an ester instead of an amide moiety suppresses a hydrogen bond and introduces a repulsive interaction between oxygen-centered lone pairs. This in turn is responsible for the 1000-fold reduction of binding affinity of vancomycin for the peptidoglycan [12,29,30].

**Figure 2 molecules-27-08927-f002:**
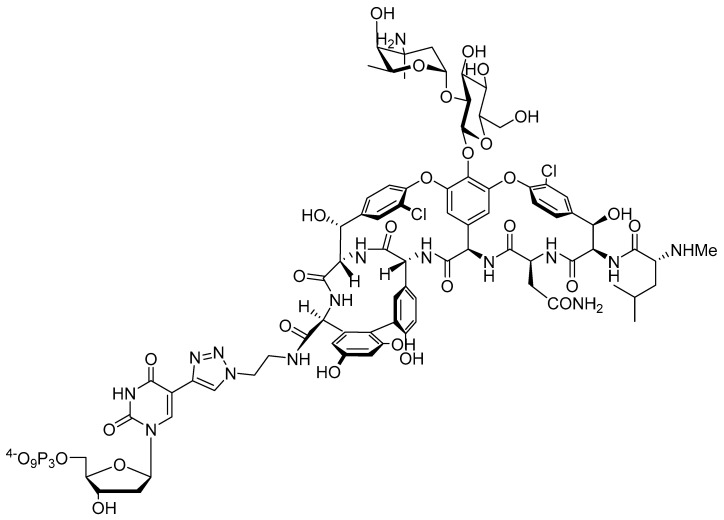
Chemical structure of modified nucleotide dU**^Van^**TP.

**Figure 3 molecules-27-08927-f003:**
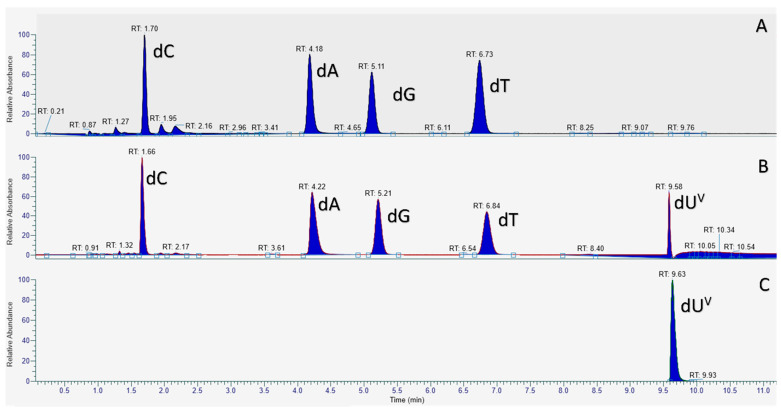
LC-MS chromatograms of deoxynucleosides from digested PEX reaction products obtained with dU**^Van^**TP and dTTP. (**A**) LC-MS chromatogram of the digested product obtained with dTTP; (**B**) LC-MS chromatogram of the digested product obtained with dU**^Van^**; (**C**) LC-MS chromatogram of the synthesized dU**^Van^** standard.

**Figure 4 molecules-27-08927-f004:**
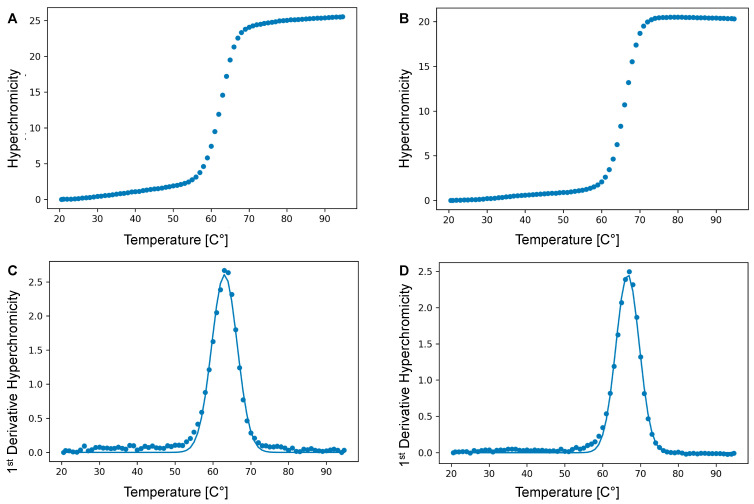
Illustrative UV melting curves (λ = 260 nm) of: (**A**) the vancomycin-modified duplex and (**B**) the corresponding natural DNA duplex. The corresponding first derivatives are shown in panels (**C**) and (**D**), respectively. The total duplex concentration was 2 µM in 137 mM NaCl, 2.7 mM KCl, 4.3 mM Na_2_HPO_4_, 1.47 mM KH_2_PO_4_, and at pH 7.4.

**Figure 5 molecules-27-08927-f005:**
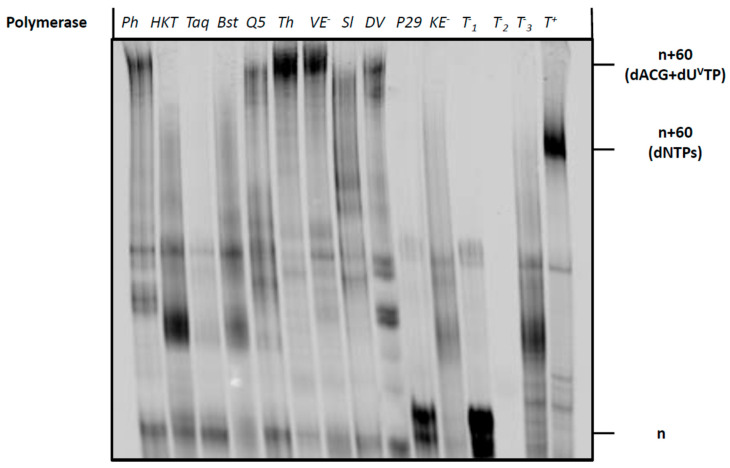
Gel analysis (PAGE 20%) of PEX reaction products carried out with the Phusion (Ph) (2U), Hemo Klem Taq (*HKT*) (8U), Taq (5U), Bst (8U), Q5 (2U), Therminator (Th) (2U), Vent *(exo*^−^*)* (VE^−^) (2U), Dpo4 (Sl) (2U), Deep Vent (DV) (2U), Phi29 (P29) (10U) and Kf *(exo*^−^*)* (*KE*^−^) (5U) DNA polymerases with primer **P2** and template **T2**. Each reaction mixture contained **T2** (15 pmol), **P2** (10 pmol), dATP, dCTP, dGTP (200 μM), dU**^Van^**TP (200 μM), polymerase buffer (1 μL for 10X and 2 μL for 5X) and the respective polymerase and was incubated for 4 h at the appropriate temperature. The negative controls were performed without polymerase (T^−^_1_), primer **P2** (T^−^_2_) and dU**^Van^**TP (T^−^_3_). The positive control (T^+^) contained dTTP instead of dU**^Van^**TP. n corresponds to unreacted primer **P2**, n+60 (dNTPs) corresponds to the expected full-length product in presence of dNTPs and n+60 (dATP, dCTP, dGTP + dU**^Van^**TP) corresponds to the expected full-length product in presence of dU**^Van^**TP instead of dTTP.

**Figure 6 molecules-27-08927-f006:**
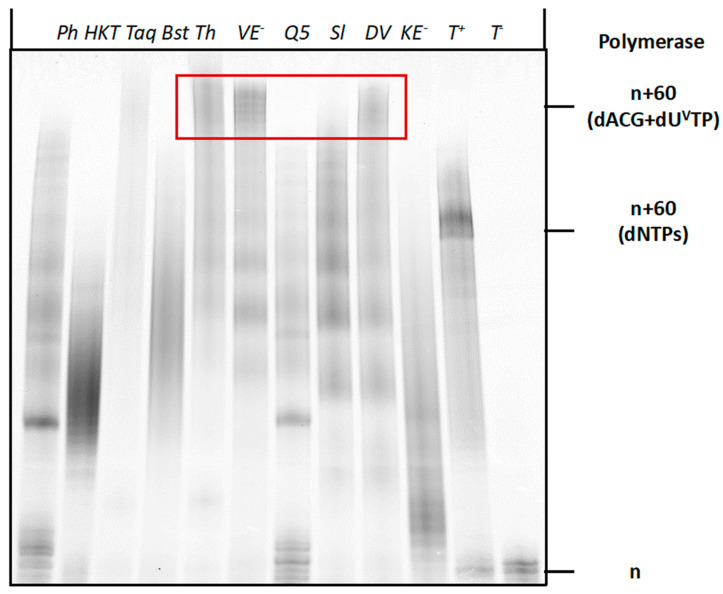
Gel analysis (PAGE 20%) of PEX reaction products carried out with the Phusion (Ph) (2U), Hemo Klem Taq (HKT) (8U), Taq (5U), Bst (8U), Q5 (2U), Therminator (Th) (2U), Vent *(exo*^−^*)* (VE^−^) (2U), Dpo4 (Sl) (2U), Deep Vent (DV) (2U), Phi29 (P29) (10U) and Kf *(exo*^−^*)* (*KE*^−^) (5U) DNA polymerases with primer **P2** and template **T3**. Each reaction mixture contained **T3** (15 pmol), **P2** (10 pmol), dATP, dCTP, dGTP (200 μM), dU**^Van^**TP (200 μM), polymerase buffer (1 μL for 10X and 2 μL for 5X) and the respective polymerase and was incubated for 4 h at the appropriate temperature. The negative controls were performed without polymerase (T^−^_1_), primer **P2** (T^−^_2_) and dU**^Van^**TP (T^−^_3_). The positive control (T^+^) contained dTTP instead of dU**^Van^**TP. n corresponds to unreacted primer **P2**, n+60 (dNTPs) corresponds to the expected full-length product in presence of dNTPs and n+60 (dATP, dCTP, dGTP + dU**^Van^**TP) corresponds to the expected full-length product in presence of dU**^Van^**TP instead of dTTP.

**Figure 7 molecules-27-08927-f007:**
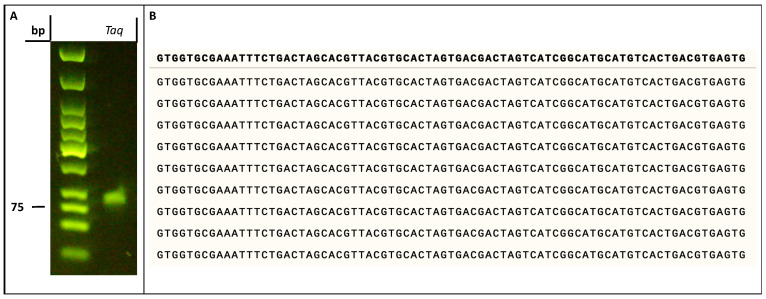
(**A**) Gel image (agarose 2%) analysis showing the product of the conversion of dU**^Van^**-modified dsDNA into natural dsDNA by means of PCR in presence of Taq as a polymerase. The reaction was performed in 25 μL and contained purified dU**^Van^**-modified dsDNA (10 nM), primers **P3**/**P4** (500 nM), dNTPs (200 μM), Mg^2+^ (2 mM), Thermopol buffer (2.5 μL of 10X) and Taq (5U, 1 μL). The PCR program was 95 °C for 5 min, (95 °C 30′’, 52 °C 30′’, 75 °C 30′’) X 10 cycles, and a final elongation at 75 °C for 5 min. (**B**) Sequence alignment of nine different plasmid colonies obtained by PCR conversion of dU**^Van^**-modified dsDNA to natural DNA followed by cloning and sequencing. The first line in bold corresponds to the sequence of template **T2**, while the following nine lines correspond to the sequenced products.

**Table 1 molecules-27-08927-t001:** DNA primer and templates used for primer extension reactions and PCR to evaluate the compatibility of dU**^Van^**TP with enzymatic DNA synthesis ^a^.

P1	5′-TAC GAC TCA CTA TAG CCT C’
**T1**	5′-**A**GA GGC TAT AGT GAG TCG TA
**T2**	5′-*CAC TCA CGT CAG TGA CAT GCA* TGC CG**A** TG**A** CT**A** GTC GTC **A**CT **A**GT GC**A** CGT **AA**C GTG CT**A** *GTC AGA AAT TTC GCA CCA C*
**T3**	5′-*CAC TCA CGT CAG TGA CAT GC* **N_40_** *GTC AGA AAT TTC GCA CCA C*
**T4**	5′-Phos-**A**GA GGC TAT AGT GAG TCG TA
**P2**	5′-FAM-GTG GTG CGA AAT TTC TGA C
**P3**	5′-GTG GTG CGA AAT TTC TGA C
**P4**	5′-CAC TCA CGT CAG TGA CAT GC

^a^ italicized letters represent primer binding sites and bold, red letters indicate sites where modified nucleotides can be incorporated. N40 represents the randomized region of the degenerate library.

## Data Availability

The data presented in this study are available on request from the corresponding author.

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
