# Peer review of "Enzymatic Synthesis of Vancomycin-Modified DNA"

_molecules, 2022, doi:10.3390/molecules27248927_

Round 1

Reviewer 1 Report

The manuscript by Marcel Hollenstein et al reported a synthesized deoxyuridine analogue containing a vancomycin residue. The nucleotide showed compatible with enzymatic DNA synthesis and SELEX methodology. It is a novel and interesting modification of vancomycin that could improve its AMR. The manuscript overall was well written which could be published after minor revisions.

1.  It would be perfect if authors can provide the antibacterial activity of modified aptamers of vancomycin, although it mentioned that the work is undergoing.

2. I would suggest citing and mentioning Prof. Dale Boger’s Vancomycin3.0 by introducing three synergistic mechanisms action for the significantly improved activity with minimal antimicrobial resistance (PNAS2017)

Author Response

Comments and Suggestions for Authors

The manuscript by Marcel Hollenstein et al reported a synthesized deoxyuridine analogue containing a vancomycin residue. The nucleotide showed compatible with enzymatic DNA synthesis and SELEX methodology. It is a novel and interesting modification of vancomycin that could improve its AMR. The manuscript overall was well written which could be published after minor revisions.

-------------------

Response: We thank this reviewer for the positive assessment of our manuscript.

  1. It would be perfect if authors can provide the antibacterial activity of modified aptamers of vancomycin, although it mentioned that the work is undergoing.

--------

Response: We thank this reviewer for this interesting suggestion. Please also refer to our answer to point #5 of reviewer #3 and our comments to reviewer#2. Evaluation of the antibacterial activity of modified aptamers will require numerous steps including cell-SELEX followed by NGS analysis after 10-15 rounds of SELEX. If the SELEX protocol leads to an efficient enrichment of the library, we will prepare enzymatically the modified sequences to evaluate their binding affinities and specificities. Potent aptamers will then be prepared by solid-phase synthesis to evaluate their antimicrobial activity. This bulk of this work is currently ongoing in our laboratories but is beyond the scope of this present manuscript. However, a recent publication using a primer equipped with a hotspot peptide in SELEX instead of a modified nucleotide highlights the feasibility of this approach. We have included the following sentence in the revised manuscript:

A similar approach using a primer equipped with a hotspot peptide in SELEX was re-cently employed for the selection of aptamers against the human angiotensin-converting enzyme 2 (hACE2) underscoring the usefulness of such a method.[76]

And inserted a citation to the following reference:

  1. Lee, M.; Kang, B.; Lee, J.; Lee, J.; Jung, S. T.; Son, C. Y.; Oh, S. S., De novo selected hACE2 mimics that integrate hotspot peptides with aptameric scaffolds for binding tolerance of SARS-CoV-2 variants. Sci. Adv. 2022, 8, (43), eabq6207

  1. I would suggest citing and mentioning Prof. Dale Boger’s Vancomycin3.0 by introducing three synergistic mechanisms action for the significantly improved activity with minimal antimicrobial resistance (PNAS2017)

------

Response: We thank the reviewer for this suggestion. We have now cited this nice work in the revised manuscript as reference#24.

Reviewer 2 Report

Journal Molecules (ISSN 1420-3049)

Manuscript ID:  molecules-2056734

TitleEnzymatic synthesis of vancomycin-modified DNA

Authors

Chiara Figazzolo , Frédéric Bonhomme , Saidbakhrom Saidjalolov , Mélanie Ethève-Quelquejeu , Marcel Hollenstein *

Abstract

Many potent antibiotics fail to treat bacterial infections due to emergence of drug-resistant strains. This surge of antimicrobial resistance (AMR) calls in for the development of alternative strategies and methods for the development of drugs with restored bactericidal activities. In this context, we surmised that identifying aptamers using nucleotides connected to antibiotics will lead to chemically modified aptameric species capable of restoring the original binding activity of the drugs and hence produce active antibiotic species that could be used to combat AMR. Here, we report the synthesis of a modified nucleoside triphosphate equipped with a vancomycin moiety on the nucleobase. We demonstrate that this nucleotide analogue is suitable for polymerase-mediated synthesis of modified DNA and importantly, highlight its compatibility with the SELEX methodology. These results pave the way for bacterial-SELEX for the identification of vancomycin-modified aptamers.

Comments

Nowadays antimicrobial resistance (AMR) is a hot topic, due to the failure of bacterial infection treatments. There is an urgent need to tackle the emergence of drug resistance. The present work describes an approach to modify with aptamers using nucleotides of a well-known and used antibiotic, like vancomycin, to avoid drug resistance and restore the original binding activity. 

The introduction section reports the state of the art and advantages of the methodology with respect to the state of the art. 

The results are clearly presented and supported by the figures for the synthesis and chemical and biochemical characterization of modified vancomycin-bearing deoxyuridine triphosphate dUVanTP. However, even though the Authors are working on the Cell-SELEX experiments against vancomycin-resistant strains, evaluating the antibacterial activity of modified aptamers would provide added value to the current work.

The quality of the figures needs to be slightly improved. On the other hand, the figures’ caption well describes the figure panel. 

The experimental section and supporting information are detailed and well-described.

The conclusion is in agreement with the results reported.

More in general the paper is well written and easy to read.

I’m glad to consider this manuscript for publication.

Author Response

Comments:

Nowadays antimicrobial resistance (AMR) is a hot topic, due to the failure of bacterial infection treatments. There is an urgent need to tackle the emergence of drug resistance. The present work describes an approach to modify with aptamers using nucleotides of a well-known and used antibiotic, like vancomycin, to avoid drug resistance and restore the original binding activity.

The introduction section reports the state of the art and advantages of the methodology with respect to the state of the art. 

The results are clearly presented and supported by the figures for the synthesis and chemical and biochemical characterization of modified vancomycin-bearing deoxyuridine triphosphate dUVanTP. However, even though the Authors are working on the Cell-SELEX experiments against vancomycin-resistant strains, evaluating the antibacterial activity of modified aptamers would provide added value to the current work.

----------------

Response: We thank this reviewer for this comment. Both reviewers #1 and #3 have made a similar comment, suggesting to include data on the antibacterial activity of resulting modified aptamers. While this would certainly further improve the quality of our manuscript, this represents a massive amount of work that is beyond the scope of the present manuscript. We believe that the work presented herein represents a first and very important step towards this aim. We demonstrate for the first time that nucleotides can be equipped with large antibiotic analogues and that the resulting modified nucleotide is not only compatible with enzymatic DNA synthesis but also with the SELEX methodology. We believe that establishing conditions for the enzymatic synthesis of DNA using antibiotic-containing nucleotides compatible with SELEX is of high appeal to any laboratory that tries to explore a similar route for the development of novel antimicrobial, aptamer-based agents. Results from the cell-SELEX, including binding affinities and specificities, and the antimicrobial activity of modified aptamers will be reported in due course.

The quality of the figures needs to be slightly improved. On the other hand, the figures’ caption well describes the figure panel.

-------------

Response: We thank this reviewer for this comment. We do agree and we have changed Figure 3 which in our opinion was of lower quality and improved Figure 1.

The experimental section and supporting information are detailed and well-described. The conclusion is in agreement with the results reported. More in general the paper is well written and easy to read. I’m glad to consider this manuscript for publication.

----------

Response: we thank this reviewer for the very positive assessment of our manuscript.

Reviewer 3 Report

Alternative strategies and methods for the development of drugs with restored bactericidal activities need to be developed because of the surge of antimicrobial resistance (AMR). The authors claimed that a deoxyuridine analogue bearing a vancomycin residue at position C5 of the nucleobase-dUVanTP was synthesized and shown to be compatible with enzymatic DNA synthesis under PEX reaction conditions, despite the presence of such a bulky chemical modification, which is expected to be the follow-up solution of vancomycin resistance of bacteria. However, some issues should to be addressed properly.

1. The manuscript should be structured according to the IMRaD format (Introduction, Methods, Results and Discussion) to meet the requirements of journal.

2. Results part should to be divided into several paragraphs and subtitled for easy reading and understanding. The Citation format of the manuscript must be done according the requirements of the journals. Figure legends should be added to the figures.

3. The title should be a concise, meaningful and easy understanded message on outcome with no more than 20 words rather than a description of approach. Please avoid using too much backgrounds or details of methodology or experimental approach in the results section. Details of methodology or experimental approach should be added to the Methods and Materials section.

4. The recognition of the aptamer highly depends on the folding structure of the initial aptamer, which may lead to changes in the folding structure of the modified aptamer, affecting its affinity and specificity for recognition. This negative effect should be discussed in the manuscript.

5. The preliminary results of Cell-SELEX experiments against vancomycin resistant strains are encouraged to be provided to the evaluate the efficiency of the method and also the antibacterial activity of modified aptamers.

Author Response

Comments and Suggestions for Authors

Alternative strategies and methods for the development of drugs with restored bactericidal activities need to be developed because of the surge of antimicrobial resistance (AMR). The authors claimed that a deoxyuridine analogue bearing a vancomycin residue at position C5 of the nucleobase-dUVanTP was synthesized and shown to be compatible with enzymatic DNA synthesis under PEX reaction conditions, despite the presence of such a bulky chemical modification, which is expected to be the follow-up solution of vancomycin resistance of bacteria. However, some issues should to be addressed properly.

-------------

Response: We thank this reviewer for evaluating our manuscript and all the important and useful suggestions and comments to improve the quality of our article.

  1. The manuscript should be structured according to the IMRaD format (Introduction, Methods, Results and Discussion) to meet the requirements of journal.

------

Response: We thank the reviewer for this suggestion. This was done by the editorial team of the journal and we have further improved this in the revised manuscript.

  1. Results part should to be divided into several paragraphs and subtitled for easy reading and understanding. The Citation format of the manuscript must be done according the requirements of the journals. Figure legends should be added to the figures.

------------

Response: We thank this reviewer for these suggestions. We had integrated subsections in the results and discussion sections but these were not clearly apparent since they were not numbered. We have corrected this mistake and included a numbering for each subsection in order to make the manuscript easier to read as suggested. The citation format was adjusted to the journal as suggested. All Figures were already described with captions in the original manuscript. Reviewer#2 actually mentions that all captions were adequate. Our manuscript was also proof-read by a native English-speaker and we believe that the level of English is now of good quality.  

  1. The title should be a concise, meaningful and easy understanded message on outcome with no more than 20 words rather than a description of approach. Please avoid using too much backgrounds or details of methodology or experimental approach in the results section. Details of methodology or experimental approach should be added to the Methods and Materials section.

------------

Response: We thank the reviewer for these comments. We believe that the title of our manuscript (Enzymatic synthesis of vancomycin-modified DNA) fits the criteria suggested by this reviewer. We have also removed some details of methodology (notably in the section describing the conversion to ssDNA) as suggested by this reviewer.

  1. The recognition of the aptamer highly depends on the folding structure of the initial aptamer, which may lead to changes in the folding structure of the modified aptamer, affecting its affinity and specificity for recognition. This negative effect should be discussed in the manuscript.

------------

Response: We thank the reviewer for this comment. Since we will use the modified nucleotide directly in the Cell-SELEX experiment, the resulting aptamer does not require any further, post-SELEX modification. If we had selected for an unmodified aptamer and then tried to include modified nucleotides and/or vancomycin residues at the 3’/5’ termini by solid-phase synthesis, a labour-intensive and uncertain post-SELEX protocol would have been required as suggested by this reviewer. To clarify that this will not be the case, we have now included the following sentence in the conclusions section:

However, the advantage of our approach is that multiple vancomycin-bearing nucleotides will be integrated within the scaffold of the aptamer and will be involved in the binding interactions. Hence, uncertain, labor-intensive post-SELEX structure-activity relationship studies to optimize aptamers will not be required.

  1. The preliminary results of Cell-SELEX experiments against vancomycin resistant strains are encouraged to be provided to the evaluate the efficiency of the method and also the antibacterial activity of modified aptamers.

------------

Response: A similar comment was made by the other two reviewers (see response to comments #1 of reviewers #1 and #2). We agree that these experiments would greatly enhance the quality of our present work. However, to be able to assess the antimicrobial activity of modified aptamers, we first need to select modified aptamers, ensure that the candidates are not only high binders but also specifically binding to the desired vancomycin-resistant bacterial targets, and synthesize larger amounts of modified aptamers. In this work, we demonstrate for the first time that nucleotides can be equipped with this bulky compound and that the resulting modified nucleoside triphosphate is compatible with enzymatic DNA synthesis and the SELEX methodology. This sets the stage for preparing vancomycin-modified aptamers without requiring a labour-intensive and uncertain post-SELEX modification. We will publish results on the antimicrobial resistance of vancomycin-modified aptamers in due time.

Round 2

Reviewer 3 Report

1.  Part 4 “Experimental Section” should be changed into “Materials and Methods”, and add a serial number for each methods.